

# Climate field reconstructions for the North Atlantic region of annual, seasonal and monthly resolution spanning CE 1241-1970

Jesper Sjolte [1] and Qin Tao [1]

[1]Department of Geology – Quaternary Science, Lund University, Sölvegatan 12, 223 62, Lund, Sweden.
**Correspondence:** Jesper Sjolte  (jesper.sjolte@geol.lu.se)

**Abstract.** The North Atlantic region is a key component of the climate system via large scale atmosphere and ocean circulation. Climate field reconstructions can provide a long-term context for ongoing climate change and contribute to our understanding of climate dynamics, impact of external forcings, and act as references for model evaluation and baseline for natural variability. There are distinct differences in North Atlantic climate variability between the seasons in terms of climate modes and amplitude
of the variance. Constraining long-term climate variability in sub-annual resolution is therefore needed for a more complete understanding of the governing processes. In this study we present reconstructed climate in annual, seasonal and seasonal resolution based on a small high-quality network of proxy data combined with output from an isotope enabled climate model. Compared to earlier work we have improved the methodology to obtain better skill across a larger area and more realistic variance of the reconstructed variables which include 2m temperature (T2m), sea surface temperature (SST), sea level pressure
(SLP) and precipitation amount. Here we validate the reconstructions against reanalysis data, observed SST and eight long-term records of observed temperature. The reconstructed temperature correlates with up to 0.71 for seasonal and 0.68 for annual data compared to reanalysis data, while the correlation is about 0.3 for monthly resolution. The skill for SLP shows the imprint of large-scale circulation for winter with more local pattern dominating for summer. This is also mirrored in the skill for precipitation. In addition, the reconstructed annual mean SST shows basin-wide skill for the North Atlantic, indicating
relevance of the reconstruction to studies of atmosphere-ocean interaction. In summary, the results show the potential of assimilating a small high-quality network of proxy records.

## 1 Introduction

The climate of the past millennium is an important reference period for current and future climate change due to relatively abundant climate proxy data and while also spanning the onset of the industrialisation of society (Jungclaus et al., 2017). De-
spite the good coverage of proxy data many challenges remain with respect to temporal resolution and spatial representation of climate reconstructions. Climate reconstructions are often representing annual mean data (e.g., Tardif et al., 2019), or only the summer season (Büntgen et al., 2020), and can be limited to one climate variable only, e.g., temperature. This is to a large extent due to the readily available tree-ring data for this period and/or uncertainties in dating and seasonality of other types of proxy data. However, key information of the spatiotemporal variability and dynamics is lost when choosing to target the annual mean
for reconstructions, for example due to seasonal differences in the main climate modes (Sjolte et al., 2020). Several different





approaches have been taken to reconstruct past climate modes of the North Atlantic region, most notably for reconstructions of the main mode of wither variability, namely the North Atlantic Oscillation (NAO). One approach is to specifically target NAO variability and reconstruct a time series of NAO (Trouet et al., 2009; Ortega et al., 2015; Michel et al., 2020), while others aim to reconstruct the pressure field and then extract the main modes (Luterbacher et al., 2004; Sjolte et al., 2018, 2020). The latter

approach has the advantage of potentially separating different climate modes such as NAO and the Eastern Atlantic pattern, and thereby avoiding to assign variability in the proxy data to the reconstructed NAO which might be due to imprints of other climate modes.

Recently, climate field reconstructions of monthly resolution have also been produced (Valler et al., 2021, 2024). For this work instrumental data and historical documents have been included, which can constrain the variability on shorter time-scales than

the seasonal resolution possible from climate proxy records. However, due to the lack of the instrumental data and historical documentation in the earlier part of the reconstructions the skill cannot be expected to be consistent from the early to the later part of the reconstructed time frame. This can for example cause data-dependent shifts in the spatial patterns of atmospheric modes (Tao et al., 2023).

The skill of climate reconstructions is invariably linked to the climate signal preserved in proxy data. Quantifiable climate

information can be found in archives preserving the variability of the isotopic composition of precipitation. In extratropical regions the mean isotopic composition of precipitation is strongly correlated with the local temperature (Dansgaard, 1964). The isotopic composition is usually formulated as the relative deviation from the Vienna mean standard ocean water composition using delta notation, e.g., $\delta^{18}O$ for the relative abundance of $^{18}O$ in a water sample (Craig, 1961). Despite the linear spatial relation between $\delta^{18}O$ and temperature, there are large regional differences in the relationship between local climate

and temporal variations in $\delta^{18}O$, which hampers a simple translation from $\delta^{18}O$ to temperature (e.g., Sjolte et al., 2011). However, $\delta^{18}O$ remains one of the most important sources of paleoclimate variability from polar ice cores (Jouzel, 2013). The availability of seasonally resolved $\delta^{18}O$ ice core data is determined by the accumulation rate and annual layer thickness. The annual cycle of $\delta^{18}O$ is attenuated due to diffusion in the firn and the annual cycle is lost at low accumulation sites (Johnsen, 1977). While the annual cycle can be partly restored using mathematical back-diffusion (Johnsen et al., 2000), these limitations

mean that only relatively few ice core sites can be used for millennium scale seasonal reconstructions.

While the aforementioned advantages of seasonal reconstructions are clear, the main limitations lie in the availability of well-dated, long-term seasonally resolved proxy data. As already mentioned above, tree-ring data ticks many of the boxes for high quality seasonal data. However, several caveats are still present when selecting tree-ring data for reconstructions. Firstly, in term of tree-ring width or maximum late wood density (MXD), not all trees are strongly sensitive to temperature (Büntgen

et al., 2008). Secondly, sampling replication of tree-ring chronologies vary over time, typically with decreasing replication of older trees, which increases the uncertainty back in time (Ljungqvist et al., 2020) The recent decades have seen an increase in studies of oxygen isotope data measured on tree ring cellulose (Balting et al., 2021). However, the number of millennium-length isotope chronologies are still quite limited, and the interpretation of the data is not straightforward due to both hydrological and biophysical influences on the isotope signal (e.g. Seftigen et al., 2011; Balting et al., 2021).

Gridded climate field reconstructions can be analysed in similar ways as climate model data or meteorological reanalysis data,





which is very useful for comparing these different datasets. A range of methods can be used to produce climate field reconstructions, with the common feature being that climate model output is resampled for the best fit to climate proxy data, or using spatial statistics to infer climate patterns (see also Smerdon et al. (2023)). Challenges for climate field reconstructions include uneven distribution of proxy data sites. This tends to concentrate both skill and variance in the areas near the proxy data sites, which skews the spatiotemporal variability and creates biases the representation of the main modes of variability (Sjolte et al., 2020; Tao et al., 2023).

In this study (hereafter, SAT25) we use seasonally resolved climate proxy data and an isotope enabled climate model to produce climate reconstructions of annual, seasonal and monthly resolution for sea level pressure, temperature and precipitation for summer and winter over the North Atlantic region covering the time period 1241-1970. The climate proxy data includes Greenland ice core $\delta^{18}$O data, tree ring MXD, blue intensity (BI), as well as tree ring cellulose $\delta^{18}$O$_{cell}$. Our study can be seen as an update to the previous reconstructions by Sjolte et al. (2018, 2020) (hereafter, SEA18 and SEA20) including expanded proxy data network and improved methodology. We evaluate the reconstructions against reanalysis data, long-term temperature observations and compare with other reconstructions.

## 2 Data

### 2.1 Proxy and climate model data

We select the proxy data with the following criteria. Firstly, the proxy data should have a strong imprint of climate. In case of the MXD, tree-ring width (TRW) and BI data from tree-ring chronologies, this means high correlation to local temperature, while we look for a clear imprint of atmospheric circulation in the $\delta^{18}$O-based records. Secondly, the proxy data needs to have 0–1-year dating uncertainty and a well-defined seasonality. To achieve consistent performance of the reconstruction we only select proxy records that span the whole time frame of the reconstruction (CE 1241-1970) (Figure 1 and Table 1).

We use the isotope enabled version of the climate model ECHAM5/MPIOM (Werner et al., 2016) run for the past 1200 years forced with natural and anthropogenic forcings. This is the same model simulation used in SEA18 and SEA20.

The atmosphere component is run in a T31L19 configuration corresponding to 3.75 x 3.75 degrees horizontal resolution and 19 vertical layers, while the ocean component, MPIOM GR30L40, is set to 3 degrees horizontal resolution with 40 layers. The model features isotope fractionation during all phase changes, including kinetic fractionation during evaporation and in mixed-phase clouds.

### 2.2 Interpretation of $\delta^{18}$O$_{cell}$

The variability of $\delta^{18}$O$_{cell}$ is governed by several different processes, some of which have a high degree of co-variability. From the following analysis we assume that the main control of $\delta^{18}$O$_{cell}$ is the isotopic composition of precipitation forming the





groundwater, which comprises the source water for formation of xylem sap used to form cellulose during the growing season. Depending on the hydrological setting at the site of the tree-ring chronology there can be a lagged isotope signal from precipitation formed prior to the growing season, or from climate driven intra-annual variability. The lag can be due to groundwater

recharge during spring melt of snow or sap formed during late winter and spring (Wang et al., 2005). Our tests indicate both $\delta^{18}O_{cell}$ records used in this study are sensitive to weather variability (SLP, T2m, precip. amount) during the growing season (Figure S1), as well as the weather variability of the winter preceding the growing season (Figure S2). The records ENGL and CZEC are mainly correlated to variability connected to the winter NAO, which can be explained by the JJA Palmer drought severity index (pdsi) at these sites being sensitive to the DJF NAO. This is due to moisture availability of the growing season

being affected by the preceding winter precipitation, as well as land-atmosphere feedback mechanisms where weather patterns promoting summer drought are enhanced by low soil moisture of the previous winter season (Wang et al., 2011). Based on these findings we use the $\delta^{18}O_{cell}$ records as representing a mixture of summer and winter variability. We use JJA for representing the growing season and the extended winter season (Nov-Apr) to represent the winter preceding the growing season. Using the extended winter accounts both for impact of the winter circulation patterns, precipitation and composition ground-

water for early spring sap formation. While a mechanistic approach to connect the $\delta^{18}O_{cell}$ to $\delta^{18}O$ in precipitation is possible through forward modelling (e.g. Roden et al., 2000) this would also introduce new parameters with uncertainties specific to each site. Since the untreated $\delta^{18}O_{cell}$ in the first place has a clear imprint of atmospheric circulation for summer and winter, we choose to assume that the variability of $\delta^{18}O_{cell}$ primarily is governed by a mix of $\delta^{18}O$ in precipitation for both seasons.

**2.3   Observations, reanalysis and reconstrcutions for validation and comparison**

We use a variety of data ranging from instrumental data to climate field reconstructions to evaluate our climate reconstruction. For assessing spatial coverage of skill, we use the 20th Century Reanalysis version 3 (20CRv3) (Slivinski et al., 2021). We compared the 20CRv3 precipitation against the CPC gridded precipitation dataset (Chen et al., 2008) and found them to perform comparably on seasonal timescales. Due to 20CRv3 also covering ocean grid points, we prefer the 20CRv3 for evaluation

of the reconstructed precipitation field, despite not being constrained directly by observed precipitation. As a supplement to evaluating the reconstruction with the 20CRv3 temperature we use long-term observed temperature from Greenland (Vinther et al., 2006), Iceland (Jónsson, 1989), England (Legg et al., 2025), Denmark (Cappelen et al., 2021) and Sweden (Moberg et al., 2002; Bergström and Moberg, 2002), to achieve a longer overlapping time interval for the evaluation. To evaluate reconstructed sea surface temperature (SST) we use three different datasets ERSSTv5 (Huang et al., 2017), COBE2 (Hirahara et al.,

2014), HadISST (Rayner et al., 2003) that have different properties due to different treatment of observational data and missing values. In the discussion we go beyond observation-based datasets and compare to the climate field reconstruction ModE-RA (Valler et al., 2024) of monthly resolution covering 1421-2008.



## 2.4 Methods

We use the analogue method to assimilate proxy data using an isotope enabled climate model. For each site we evaluate the model output against the proxy data. We use different evaluation of the model data for the summer and winter season. For the summer season we use a $\chi^2$-measure for the goodness of fit:

$$\chi^2_{sum}(t) = \sum_{i=1}^{n} \frac{(m(n,t') - p(n,t))^2}{\sigma_{mp}} \tag{1}$$

Where, m is the modelled anomaly and p is the proxy anomaly at a given site, n is the number of proxies, t is the year of
the reconstruction, t´ is the year of the model run, and $\sigma_{mp}$ is the error estimated from the combined standard deviation of the model and proxy data.

The imprint of large-scale circulation on the isotopic composition is very strong for winter and the amplitude of the $\delta^{18}O$ anomalies bears a signal in it-self. Part of this signal is the gradient between proxy sites (Sjolte et al., 2018). Since we are interested in preserving as much of the signal in the proxy data as possible, we do not normalize the data when fitting the
model data to the proxy data. The measure for goodness of fit for winter is then:

$$\chi^2_{win}(t) = \sum_{i=1}^{n} (m(n,t') - p(n,t))^2 \tag{2}$$

Where, m is the modelled anomaly and p is the proxy anomaly at a given site, n is the number of proxies, t is the year of the reconstruction and t´ is the year of the model run.

For the annual mean and monthly reconstruction, the proxies for summer and winter are matched simultaneously using Eq. 1.
The monthly data from the best matching model years are then extracted, taking into account that the ice core data for winter are representing months Nov-Apr when the annual mean for the calendar mean is calculated (i.e., Nov-Dec are assigned to previous calendar year compared to the matched winter proxy data centred in January).

In this study we calculated the ensemble mean using a logarithmic weighting function. Due to the bias-variance trade off a high number of ensemble member reduces noise and increases skill in term of correlation but causes decrease in variability, as
well as skewing the variability towards the main mode and causing mixed secondary modes in the SLP field. Before arriving at our current approach, we tested different ways calculate the ensemble mean: constant number of ensemble members with equal weighting (as SEA18/20), variable number of ensemble members based on match with proxy data, also in combination with a minimum number of ensemble members. Here we calculate the weighted mean based on the $\chi^2$ fit (Eq. 1 & 2), which yields a smoother behaviour of reconstruction with less year-to-year difference in the statistical properties of the reconstruc-
tion compared to a variable number of ensemble members, while still allowing the best matching models years to have the most weight. The weighting is calculated as the logarithm of the normalized $\chi^2$-distance, where the least important ensemble member receives zero weighting (Figure S3). The $\chi^2$-distance as a function of number of ensemble members is similar to an exponential decay function and apply the logarithm to achieve a more equal weighting of the ensemble members. Otherwise,





the result would be skewed too much towards a few of the best fitting ensemble members, which results in more noise in the
reconstruction. With the applied weighting 50% of the weight goes to the best matching 30% of the ensemble members.
Time series are shown as anomalies from the mean of the full-length data series, and monthly data are anomalies with respect
to the mean annual cycle.

## 3 Results

### 3.1 Ensemble reconstruction and initial evaluation

Before comparing to observed climate we evaluate the coherence between the reconstruction and the assimilated proxy data,
and assess how many model analogues to use in the ensemble reconstruction. When determining number of ensemble members
to include for each year there are several factors to consider. A high number of ensemble members reduces noise by smoothing
spatially and temporally, but this also reduces and skews variability to the main modes of variability (e.g., NAO). Conversely, a
low number of ensemble members fail to span both the imperfect fit of the model to the proxy data, as well as the confounding
noise of the proxy data.

We first consider the match between the reconstructed signal at the sites of the proxy data. There is high correlation between
the reconstruction and all proxy records, with a match to ice core $\delta^{18}O$ and $\delta^{18}O_{cell}$ in the range r $\sim$ 0.65-0.9, and on average
slightly better correlation between reconstructed temperature and conventional tree-ring records, though in similar range (r $\sim$
0.6-0.9). We tested the match with proxy data with increasing number of ensemble members. The maximum correlation to
proxy data is achieved with 10-100 ensemble members, however the match to the proxy data only degrades slowly with up to
300 ensemble members due to the weighting with respect to $\chi^2$-distance to the proxy data (Figure S4). Using 300 ensemble
members introduces spatial smoothing which filters out noise in particularly for areas far from the proxy sites (see Section
3.4). With this high number of ensemble members, we on one hand lose information on modes of atmospheric circulation,
as mentioned in the introduction, but on the other hand gain skill for basin-wide variability for the North Atlantic region. We
175 therefore provide two versions of the reconstruction, one using 150 ensemble members and one using 300 ensemble members.
For simplicity we focus on the version with 300 emsemble members as the differences are subtle and will be explored in future
work.

Previous seasonal reconstructions (SEA18/20) only used ice core data, or introduced tree-ring data after a pre-selection of
model analogues had been made using ice core data. Here we fully incorporate the tree-ring data. The match with ice core
180 records can be achieved regardless if the tree-ring data is included or not, indicating that the analogue sampling pool is suf-
ficiently large to provide relevant analogues, and that further constraining the climate variability with records in Europe does
not hinder a good match over Greenland.

The uneven distribution of proxy data concentrates skill and variance near the sites of the proxy data, and calculating the ensem-
ble mean dampens the overall variance. We bias correct the variance of the reconstruction by rescaling the standard deviation
185 of each grid point with the corresponding standard deviation of the ECHAM5-wiso/MPIOM run. This very simple correction
improves the reconstructed variance compared to the variance for SLP, T2m and precipitation from 20CRv3 (Figures S5 &



S6).

To test the long-term stability of the reconstruction we investigated temporal variations of the match with proxy data. The average fit of model analogues is fairly constant over time, indicating that the quality of the climate reconstruction is consistent throughout 1241-1970. One reason for this consistency is the strict selection of proxy records that span the whole reconstruction, which means that that there are no methodological differences throughout the reconstruction. We found no clear preference with respect to selection of model analogues throughout the reconstruction. This is coeval with results for earlier versions of the reconstruction (SEA18/20).

## 3.2 Comparison to meteorological reanalysis

Compared to the JJA and DJF mean for 20CRv3, the reconstruction shows coherent patterns of skill for SLP and T2m, while the skill for precipitation is patchier (Figure 2). Standout features are widespread significant correlation for the winter SLP and strong correlation over the eastern North Atlantic and Northern Europe for summer T2m (max. r = 0.71). In winter the level of correlation is tightly connected with the skill for reconstructed circulation patterns. This is especially clear for the winter SLP and T2m which resemble the spatial pattern of NAO-type variability, while summer SLP and T2m which resemble the spatial pattern of Scandinavian blocking-type variability.

The correlation patterns for the annual mean are, not surprisingly, a combination of features seen for summer and winter (Figure 3 a-c). While it is less pronounced than for DJF, the signature of NAO-type variability is still evident in the spatial patterns of the correlation. The maximum correlation (r = 0.68) is achieved for T2m with lower correlations for SLP and precipitation than for winter, indicating that auto-correlation in temperature aids higher skill in some areas, while the seasonal information is lost in the annual mean SLP and precipitation due to shifting circulation patterns and different impact of circulation on T2m and precipitation on a seasonal scale.

On monthly time scales the skill of the reconstructing is lower with at maximum correlation of 0.34 for T2m (average for all months) (Figure 3 d-f). Despite lower skill, the correlation is still significant over large areas, due to the larger sampling pool (12 times as more sample points compared to annual data). The spatial patters of the correlation are similar to that of the annual mean reconstruction. Reviewing the skill for individual months, correlations peak in February and August for T2m with maximum correlation of 0.55 and 0.70, respectively (Figure S7). For SLP the imprint of large-scale circulation is again evident during winter months (max. corr. = 0.38 for February) with lower and more localized skill during summer.

## 3.3 Comparison to long-term temperature records

In addition to the comparison with 20CRv3 we compare to observed temperature from Greenland, Iceland, Central England, Denmark, and Sweden which are among the longest observed high-quality temperature records and span our study area well. For summer the high skill of the reconstruction seen in Fig. 2b is confirmed for sites in Sweden and England (Figure 4). It appears that there are periods of over and underestimated reconstructed temperature which could be a sign of underestimated





multi-decadal variability, or it could be due to a different long-term trend in the reconstruction. However, for Uppsala, which close by Stockholm and has a longer observational record, the reconstruction shows the highest correlation to observed summer temperature (r = 0.56). It is also reaffirming that the reconstruction shows good correlation to the longest existing instrumental record, the Central England temperature (r = 0.5). The lower skill for Greenland sites during summer is similar to earlier studies, because Greenland ice core summer $\delta^{18}$O is more sensitive to climate variability east of Greenland, also illustrated by

good correlation to Iceland summer temperature (r = 0.54) (Vinther et al., 2010; Sjolte et al., 2020).

The highest skill during winter is seen for Greenland winter temperatures (r ~ 0.6), although we also see good performance for Stockholm (r = 0.46), despite having no proxy site nearby (Figure 5). Compared to summer there are only the two $\delta^{18}$O$_{cell}$ proxy sites available for the winter season besides the Greenland ice cores. However, this is partly counteracted by the stronger impact of large-scale circulation on regional climate variability enabling good skill also in the Eastern part of the study area.

We observe that the variance for DJF is approximately double of the JJA variance. This is captured well for most sites by the reconstruction, also showing that the variance correction is effective also compared to instrumental data.

For annual mean data the correlation is more consistently in the range of 0.5-0.6 (Table 2), again, combining the characteristics of the summer and winter season, as well as capturing amplitude of the temperature variance (Figure S8).

The monthly reconstruction has correlations of ~ 0.3 at all sites but for Central England which has lower correlation (r = 0.21).

The lower skill for Central England is likely due to the winter variability being less coupled to the NAO. As the reconstruction is not constrained at monthly resolution, we cannot expect high skill on a month-to-month basis. What we do see is the reconstruction capturing part of the modulation of the annual cycle seen in the observations. For example, for the cold winters during the 16th and 17th century in Scandinavia (Figure S9).

## 3.4 Annual mean SST evaluation

The annual reconstruction, which targets the calendar year, enables direct comparison to more common annual reconstructions and is useful for analysing indices of sea surface temperature which are often defined on annual mean data. Our reconstruction uses a high number of ensemble members (300, ~ 25% of the available model analogues) introducing spatial smoothing to filter out noise in particular for areas far from the proxy sites. Using this high number of ensemble members, we on one hand

lose information on modes of atmospheric circulation, as mentioned in the introduction, but on the other hand gain skill for basin-wide variability for the North Atlantic region (Figure 6).

## 4 Dicussion and conclusions

Compared to the SEA18/20 reconstructions our new reconstruction shows several clear improvements in the performance for summer and winter. This includes more wide-spread skill for all reconstructed variables and more realistic variance due to

a simple, but effective, variance correction. With the introduction of the variance correction, the performance of the reconstruction is significantly improved in comparison with the 20CRv3 and observed temperature records. This indicates that the




reconstruction will lend itself well to studies of forcing attribution, climate extremes and model evaluation given the more realistic amplitude of changes featured in the reconstruction. Furthermore, we included reconstructions in annual and monthly resolution, which can be useful in comparison to other data sets. Finally, we also included reconstructed precipitation as an output variable.

There are only few reconstructions of monthly resolution spanning more than a few centuries. In the validation of our reconstruction, we see a consistent match to the proxy data, as well as the long-term observed temperature. In extension of this we have also compared our reconstruction to the ModE-RA reanalysis reaching back to 1421. In addition to proxy data ModE-RA is assimilated based on instrumental data and historical documents. Comparing our reconstruction and ModE-RA temperature for the full overlapping time period 1421-1970 we see a fair correlation for the summer months, showing large areas of coherent correlation over the North Atlantic (Figure 7). In contrast, the winter months show little in common between the two reconstructions. For SLP the coherence between ModE-RA and our reconstruction is generally low (not shown). Looking at different time periods it is clear that we find a better correspondence with the ModE-RA reanalysis our reconstruction towards the later part of the reconstruction (e.g., 20th century), where ModE-RA is incorporating instrumental data (not shown). This is especially true for the winter season, for both temperature and SLP.

Reconstructing sub-annual variability remains challenging. This is to a large extend due to the lack of available proxy data for constraining the winter season. However, as shown in SEA18/20 even a low number of high-quality proxy records is sufficient to capture large-scale winter circulation if the method is optimized for this purpose. The winter variability of SEA25 is well constrained by the ice core $\delta^{18}$O and $\delta^{18}$O$_{cell}$ throughout the reconstructed time frame.

The variability of the monthly data can also depend on how the model analogues are sampled. As explained in the Methods section, the model data for the monthly reconstruction is sampled based on the annual reconstruction. This means that the summer and winter proxies are evaluated simultaneously. We tested extracting monthly data based on the selection of model years from the seasonal reconstruction. This made some slight differences where the SLP for winter months was slightly better based on the seasonal selection of model analogues, while the temperature was slightly better based on the annual selection (Figure S7). However, if indeed targeting seasonal means we do recommend using the JJA and DJF reconstructions.

One of the novel aspects of this work is the inclusion of $\delta^{18}$O$_{cell}$ data from tree-ring chronologies. As summarized in the Introduction and Data sections, the signal recorded in $\delta^{18}$O$_{cell}$ is the result of complex biophysical processes and environmental variability. We choose not to attempt to include such complexity, and instead rely on empirical statistical relationships between the $\delta^{18}$O$_{cell}$ signal and climate variables derived from observations. In our tests of the reconstruction the inclusion of the $\delta^{18}$O$_{cell}$ data is important for the performance of the reconstruction, for both summer and winter. It provides, so to speak, the European end of the NAO see-saw which then also constrains the variability of other weather patterns. While this shows that the $\delta^{18}$O$_{cell}$ data does contain important climate information even if it is based on simple assumptions, it does not exclude that more information could be obtained using forward modelling of the biophysical and hydrological processes.

The relatively strict selection of proxy data for this work means that we have a low number of proxy sites. New proxy data can be included in future versions of the reconstruction as it becomes available. We see a need for new millennium-length $\delta^{18}$O$_{cell}$ records from Europe to constrain the winter variability further. In addition, using a simulation with an updated isotope



enabled model with better spatial resolution for the data assimilation could lead to generally improved performance of the reconstruction.

*Data availability.* The SEA25 reconstructions are archived at Zenodo (10.5281/zenodo.15746008).

290 *Author contributions.* JS initiated and designed the research project. Method development, investigation and evaluation of reconstructions were done in collaboration by JS and QT. JS wrote the first draft of the manuscript which was revised after comments by QT.

*Competing interests.* The authors declare no competing interests.

*Acknowledgements.* This work was supported by the Swedish Energy Agency (Project No. 51375-1) and the strategic research program of ModElling the Regional and Global Earth system (MERGE) hosted by the Faculty of Science at Lund University.



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



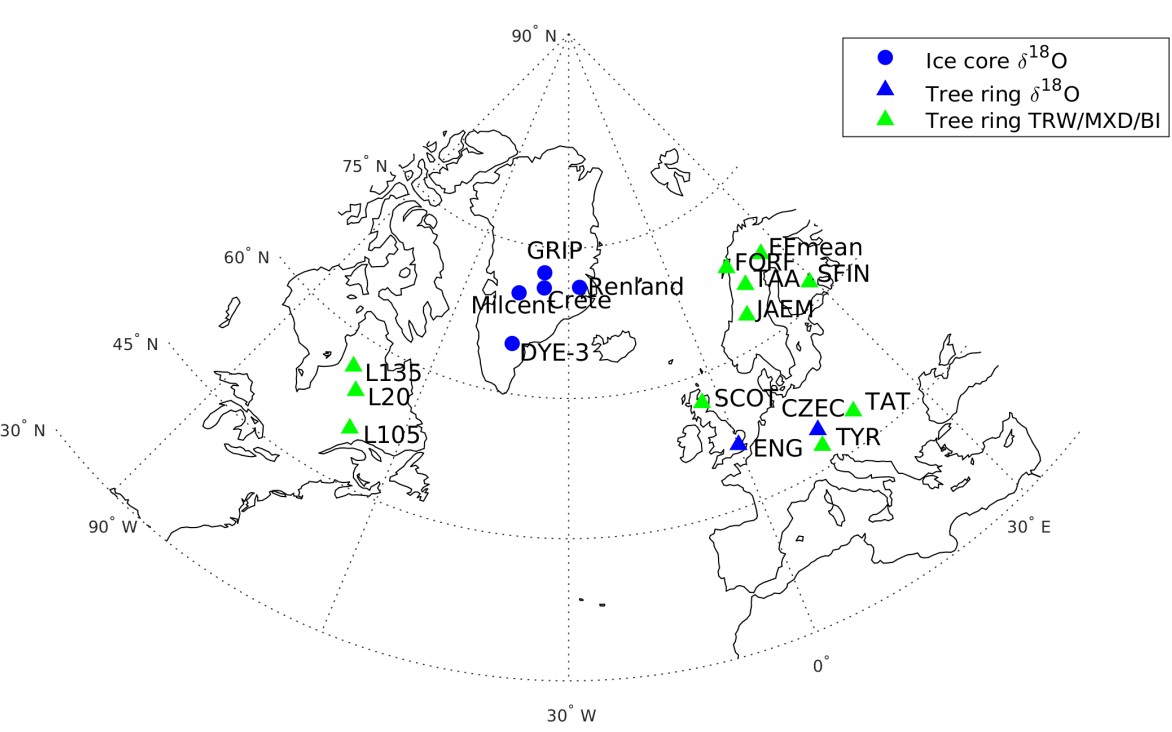

**Figure 1.** Sites for proxy data assimilated in the reconstruction. Geographical coordinates of sites and sources of data are listed in Table 1.





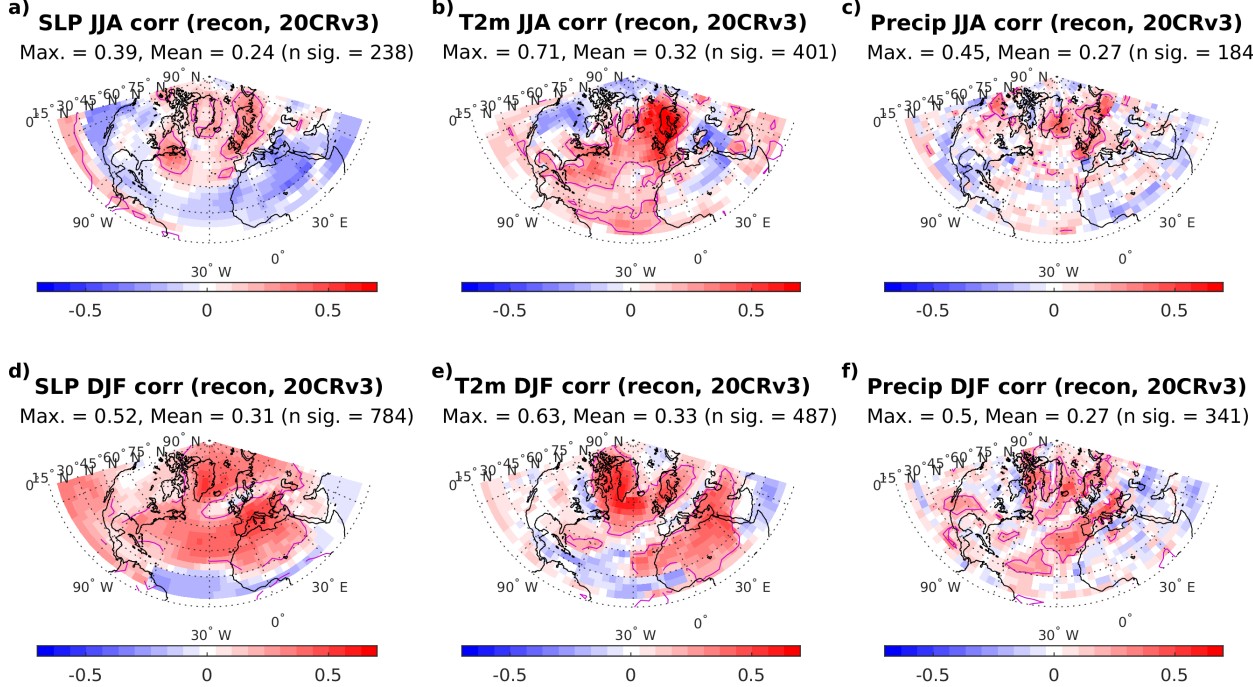

**Figure 2.** Point-wise correlation between reconstructed and 20CRv3 SLP, T2m and precipitation for JJA mean (a-c) and DJF mean (d-f) data. The maximum correlation, mean significant correlation and number of grid points with significant correlation in indicated for each subplot. Contour indicates p = 0.01.





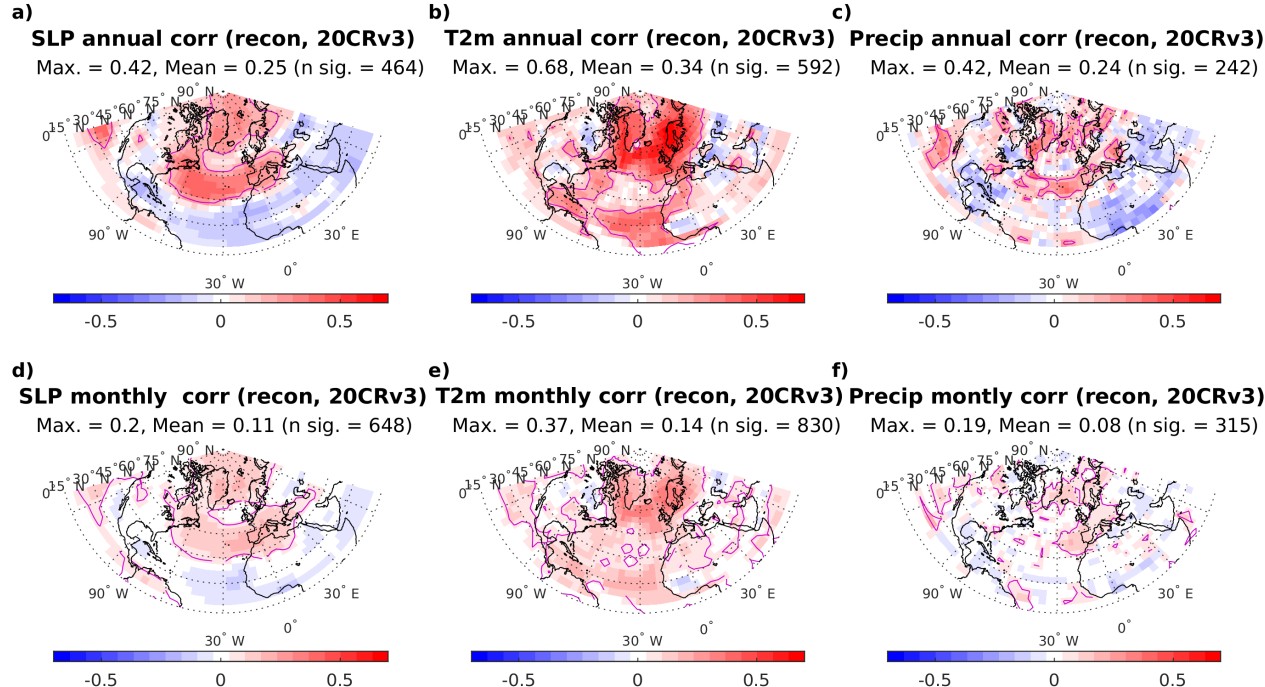

**Figure 3.** Point-wise correlation between reconstructed and 20CRv3 SLP, T2m and precipitation for annual mean (a-c) and monthly mean (d-f) data. The maximum correlation, mean significant correlation and number of grid points with significant correlation in indicated for each subplot. Contour indicates p = 0.01.



**Figure 4.** Time series of reconstructed (blue) and observed (yellow) JJA temperature for Nuuk, Ilulissat, Qaqortoq, Stykkisholmur, Central England, Copenhagen, Stockholm, and Uppsala. The blue shading indicated +/- I std dev of the ensemble reconstructed temperature.







**Figure 5.** Time series of reconstructed (blue) and observed (yellow) DJF temperature for Nuuk, Ilulissat, Qaqortoq, Stykkisholmur, Central England, Copenhagen, Stockholm, and Uppsala. The blue shading indicated +/- I std dev of the ensemble reconstructed temperature.





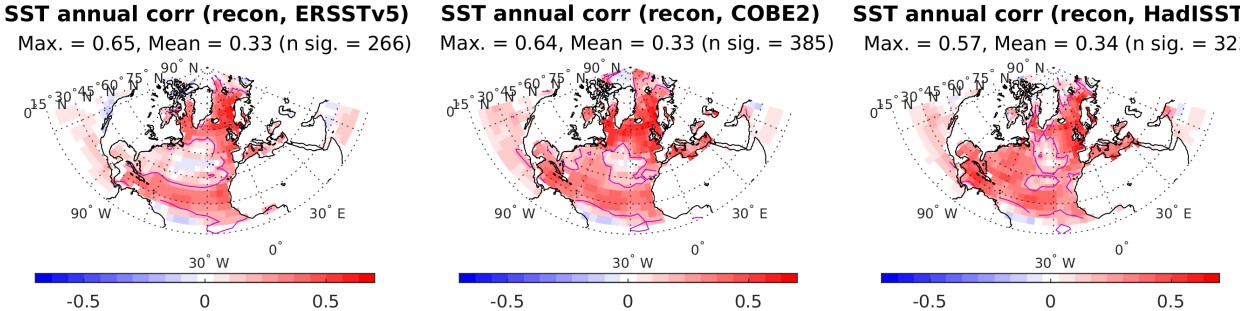

**Figure 6.** Reconstruction of annual sea surface temperature (SST) correlated point-wise to tree different SST datasets based on observations (ERSSTv5 1870-1970, COBE2 1850-1970, HadISST 1854-1970). Contour indicates p = 0.01.



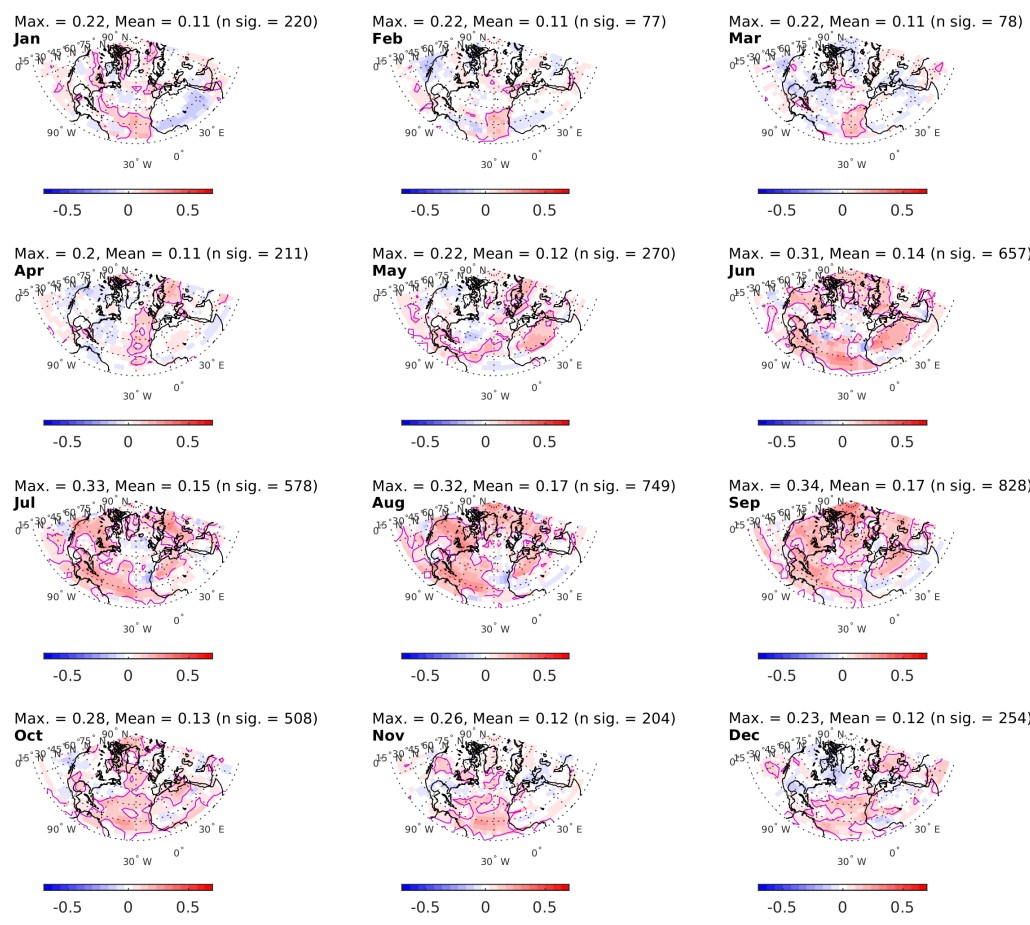

**Figure 7.** Point-wise correlation between SAT25 and ModE-RA monthly T2m. The maximum correlation, mean significant correlation and number of grid points with significant correlation in indicated for each subplot. Contour indicates p = 0.01.



**Table 1.** Proxy data sites assimilated in the reconstrution with coordinates and data sources listed.

| Location | Site name | Lat. ($^o$N) | Long. ($^o$E) | Source |
|---|---|---|---|---|
| Scotland | SCOT | 57.08 | -3.44 | (Wilson et al., 2016) |
| E Alps - Tyrol | TYR | 47.30 | 12.30 | (Wilson et al., 2016) |
| Jaemtland | JAEM | 63.30 | 13.25 | (Wilson et al., 2016) |
| Tjeggelvas, Arjeplog, Ammarnäs composite | TAA | 65.54-66.36 | 16.06-18.12 | (Wilson et al., 2016) |
| North Fenno | EFmean | 66-69 | 19-32 | (Wilson et al., 2016) |
| Forfjorddalen | FORF | 68.47 | 15.43 | (Wilson et al., 2016) |
| Tatra | TAT | 48-49 | 19-20 | (Wilson et al., 2016) |
| South Finland | SFIN | 62 | 29 | (Wilson et al., 2016) |
| Quebec | L105 | 50.80 | -68.8 | (Wang et al., 2022) |
| Quebec | L20 | 54.60 | -71.2 | (Wang et al., 2022) |
| Quebec | L135 | 56.70 | -74 | (Wang et al., 2022) |
| Cent. England | ENG | 51.5 | -1.035 | (Loader et al., 2020) |
| Central Europe | CZEC | 49.0 | 13.0 | (Büntgen et al., 2021) |
| Greenland | Crete | 71.12 | -37.32 | (Vinther et al., 2010) |
| Greenland | DYE-3 71 | 65.18 | -43.83 | (Vinther et al., 2010) |
| Greenland | DYE-3 79 | 65.18 | -43.83 | (Vinther et al., 2010) |
| Greenland | GRIP 89-1 | 72.58 | -37.64 | (Vinther et al., 2010) |
| Greenland | GRIP 89-3 | 72.58 | -37.6 | (Vinther et al., 2010) |
| Greenland | GRIP 93 | 72.58 | -37.64 | (Vinther et al., 2010) |
| Greenland | Milcent | 70.30 | -44.50 | (Vinther et al., 2010) |
| Greenland | Renland | 71.27 | -26.73 | (Vinther et al., 2010) |



**Table 2.** Correlation to Long-term temperature observations Annual, JJA, DJF, monthly. Boldface marks significant correlations with $p <$ 0.01.

| Site | Annual | JJA | DJF | monthly |
|------|--------|------|------|---------|
| Nuuk | **0.63** | 0.21 | **0.62** | **0.30** |
| Ilulissat | **0.62** | 0.13 | **0.61** | **0.31** |
| Qaqortoq | **0.55** | 0.16 | **0.63** | **0.34** |
| Stykkishólmur | **0.58** | **0.54** | **0.33** | **0.28** |
| Central England | **0.49** | **0.50** | **0.24** | **0.21** |
| Copenhagen | **0.55** | **0.55** | **0.42** | **0.31** |
| Stockholm | **0.53** | **0.48** | **0.46** | **0.32** |
| Uppsala | **0.45** | **0.56** | **0.39** | **0.27** |