# Peer review of "Climate field reconstructions for the North Atlantic region of annual, seasonal and monthly resolution spanning CE 1241-1970"

_EGUsphere, 2025_

## Author Comment (AC1)

This study presents new climate field reconstructions for the North Atlantic region spanning from 1241 to 1970 CE at annual, seasonal, and monthly resolutions, using a small network of proxy data combined with isotope-enabled climate model simulations. The authors reconstructed four key climate variables: 2-meter temperature, sea surface temperature, sea level pressure, and precipitation. Validation against reanalysis data and long-term temperature observations shows strong correlations of up to 0.7 for seasonal and annual temperature data, though monthly resolution correlations were weak. The reconstructions successfully captured large-scale winter circulation patterns in sea level pressure and demonstrated basin-wide skill for North Atlantic sea surface temperatures, providing valuable long-term climate context for understanding natural variability.

**Scientific Significance:**

This manuscript makes a meaningful contribution to paleoclimate reconstruction by extending North Atlantic climate records back to 1241 CE with multi-resolution temporal coverage. While the geographic focus and general reconstruction approach build on established methods, the improved methodology, proxy data and sub-annual resolution data represent valuable advances.

**Scientific Quality:**

The scientific approach appears methodologically sound. The validation strategy is comprehensive, employing multiple independent datasets including reanalysis data, observed SST, and instrumental temperature records. However, at least for the monthly reconstruction I would suggest a validation with instrumental data sets for the 20th century.

**Presentation Quality:**

While the abstract provides a clear overview and the introduction is very well written, some parts of the main paper could be improved. For instance, proxy selection criteria could be better explained. Results and discussion could be separated better, e.g. ModE-RA results only appear in the discussion instead of the results.

We thank the reviewer for the positive review, thorough reading of our manuscript and constructive comments. We appreciate this opportunity to improve our work based on the comments by the reviewer.

For the revision we will move the ModE-RA comparison to the results section as suggested by the reviewer.

**Specific Comments:**

1. I'm really skeptical about the monthly reconstruction in this study. ModE-RA winter temperatures at least in Europe should be well constrained by historical information and early instrumental measurements. How would Fig. 7 look for 20CR? Maybe that would be a figure for the supplement. Please also check the monthly reconstruction with gridded instrumental data sets for the 20th century. Maybe rather remove the

monthly reconstruction from the paper and "monthly" from the title or discuss it more carefully.

I our approach we choose to only include proxy data that span the entire reconstructed timeframe. We do this to have a consistent product with, in theory, same performance, throughout the reconstruction. As our method doesn't include tuning to observations, the skill obtained in the evaluation with instrumental records in principle indicates the level of performance for any time frame of the reconstruction.

The skill for monthly the reconstruction varies regionally as due to the seasonality of the proxy data and seasonal shifts in climate patterns, which is part of the reason the average monthly correlations remain low, although they peak above 0.6 for temperature during July and August (Figure S9). So, winter months and summer months do not have high skill at the same location, and the mean for all months becomes comparably low. As mentioned at the end of section 3.3, the monthly reconstruction captures a large part of the modulation of the annual cycle, and as mentioned in L253-254 the monthly reconstruction can also be useful for comparison to other datasets as you can extract any months for comparing to records with different seasonality.

We will revise the results section and discussion to clarify the text and include the explanation of the above points.

The information on the monthly comparison to 20CRv3 is actually included in the supplementary (Figure 9S), although in a somewhat convoluted format. We agree that showing monthly correlation maps for T2m would be helpful to the reader, and we will include this as a new figure in the supplementary (Figure R1 attached below).

2. How can you deal with 1 year dating uncertainty (line 80)? And how may wrongly dated proxies influence your entire study and data set? Did you consider comparing neighboring proxies with different lags to check for potential dating problems?

0-1 year dating error referred to in L80 are the conditions for choosing the proxy data. The dating of the dendrochronologies is considered to have zero uncertainty, while the ice core records are estimated to have 0–1-year uncertainty for the reconstructed period (Vinther et al., 2006). As the ice core records are dated using a combination of layer counting and volcanic horizons an entire ice core record is never shifted by one year. However, one cannot exclude that some records are off by up to one year in some intervals between dating horizons. While it goes beyond the scope of this study with such details of the ice core dating, we are happy add text to explain this in the data section.

3. Why do you "use JJA for representing the growing season and the extended winter season (Nov-Apr) to represent the winter preceding the growing season"? In this setup May does not play any role. Why not also an extended summer growing season starting in May?

This is based on investigations of the seasonal signal in the d18Ocell, where correlation to climate variables drops in spring and picks up again in the main growing season (Büntgen et al., 2021). Also, since we use the extended winter for the ice core data, the choice of the

extended winter means that we are matching with the same model d18O field for all proxy data for the winter season. This explanation will be included in the revision.

4. With this analog method you disturb the temporal evolution of the model simulations. Are the transitions from one year to the next year smooth in the ocean with its memory/autocorrelation? Especially in the monthly reconstruction I would imagine that this could be an issue.

The auto-correlation of the reconstruction is determined by the properties of the proxy data. Since the proxies are sensitive to atmospheric processes the reviewer is correct in the assumption that this results in the variance of the SST to be overestimated, although this is partly compensated by the variance correction. We will include this point in the discussion.

5. In the results you ask the question "how many model analogues to use in the ensemble reconstruction". But in the methods section above you wrote that you "calculated the ensemble mean using a logarithmic weighting function". I understood that all members are included. Maybe, I just misunderstood something but please explain this more clearly.

We assume that the reviewer is referring to L161 in the results section. It should be "how many model analogues *per reconstruction year ...*", which we will correct. So, all model years are included in the entire reconstruction, but either 150 or 300 model analogues are used per year for the two different versions of the reconstruction.

6. Fig. 2: the JJA SLP reconstruction has negative correlations if you just move slightly out of the region covered by the proxy data. What could be the reason and would you consider shrinking the reconstruction region to the area covered by proxy data?

Negative correlations can have multiple sources. For JJA the reason is most likely a combination of less well constrained large-scale circulation during summer and model biases in the JJA SLP of ECHAM5-MIPOM (Sjolte et al., 2020). We will add this to the description of the results for summer SLP. Since the temperature shows large-scale coherent patterns in correlation, we would lose this if we limit the area.

**Small remarks:**

Thank you for the corrections listed below. We will include all of them in the revision.

**Abstract:**

Line 6: "seasonal and seasonal" should probably be "seasonal and monthly"

**Introduction:**

Line 19: "... data and while ... "should be without "and"

Line 27: "the main mode of wither variability" should be "winter variability"

Data Section:

Line 110: "reconstructions" should be "reconstructions"

Line 176: "emsemble members" should be "ensemble members"

**Unclear English:**

*Introduction:*

Line 51-52: "Well-dated, long-term seasonally resolved proxy data" maybe better "well-dated, seasonally resolved, long-term proxy data"?

Line 65: "creates biases the representation" should be "creates biases in the representation"?

Results Section:

Line 193: "This is coeval with results" - "coeval" is uncommon; "consistent with" would be clearer

Line 208: "skill of the reconstructing" should be "skill of the reconstruction"

Discussion:

Line 248: "Compared to the SEA18/20 reconstructions our new reconstruction" missing comma after "reconstructions"

Line 267-269: "as shown in SEA18/20 even a low number" missing comma after "SEA18/20"

**New figure for revised supplementary**

Figure R1 Point-wise correlation between SAT25 and 20CRv3 monthly T2m. The maximum correlation, mean significant correlation and number of grid points with significant correlation in indicated for each subplot. Contour indicates p = 0.01.

---

## Author Comment (AC2)

**Summary:**

A small number of climate-sensitive proxies are used with a simulation from an isotope-enabled coupled Earth system in a field reconstruction of North Atlantic and European climate. The reconstruction spans annual, seasonal, and monthly timescales depending on the climate variable. The reconstruction is validated against instrumental products, including 20CRv3 for the atmospheric variables, and a suite of SST reconstructions for the ocean. Agreement depends on the climate field and the time period, with the best agreement in locations with abundant high-quality proxies (e.g., summer temperature in regions with latewood density records). Skill at monthly timescales, and especially for precipitation and mean-sea-level pressure, is generally poor.

There are two main issues with this study, concerning insufficient description of the method and weak validation, that need to be cleared before I can recommend publication. I elaborate on these points below and then give minor comments.

Thank you for the thorough review and constructive comments. We appreciate this opportunity to improve our work and clarify the writing in our manuscript.

**Main Points:**

1) I've read the method section twice, and I would be unable to reproduce the results in this study. The method uses a weighted-analog approach from a single model simulation. The weights are determined by misfits to the proxy records, but I don't see anything about how this comparison is made when the proxy is not a climate-model variable (e.g., MXD and ring width). This suggests that these records have been inverted for temperature, but perhaps they all have (including d180)? This is very important as it affects not only the weights, but the relevance of the validation process.

Thank you for pointing out this oversight. The tree-ring (MXD, TRW, BI) data are matched with model JJA T2m. All isotope records are matched with precipitation weighted model d18O. This is the same approach as Sjolte et al. 2020, but this should of course be clear in this paper as well. We will include a table in the revision to summarize this information.

The ad hoc variation inflation method has no justification other than the reconstructed time series has more variance. Lacking any evidence, I assume that this simply adds noise and not signal; i.e., it serves no purpose.

The inflation neither adds noise nor signal if we view it in terms of correlation. We agree that the motivation of the variance correction can be clarified in the manuscript. The variance of the model (ECHAM5-MPIOM) is the variance of the sample prior, which is the best estimate of the variance we can obtain when using this model for the assimilation. In Figure S5 and S6 we show that the variance corrected reconstruction performs well compared to reanalysis data. We will include this explanation in the revision.

2) The validation of the results is particularly weak. If, as I suspect, that many proxy records have been inverted for temperature, then those inversions have been

calibrated on the instrumental record. If true, then the validation is in sample, and doesn't independently measure skill. If I have that wrong, then this should serve as motivation to describe this aspect very clearly.

The reconstruction is not calibrated. Our method is only based on matching the model output with proxy data. As shown in Eq. 1 the data used for summer are normalized, and no tuning to observations is involved. For winter we only use d18O-based proxies, and match the model d18O with ice core and tree ring cellulose d18O (not normalized, as explained L132-135). Consequently, the validation against observed datasets (reanalysis, long-term temperature records and SST compilations) serve as a truly independent measure of skill. We will clarify this point in the revision.

Given how computationally cheap the reconstruction method is, another important validation approach is to leave out proxies for validation, and repeat the reconstruction process. This can take the form of a jackknife, or leaving out sets of proxies. The reconstruction can then be used to predict the proxies that were left out as a way to independently validate the results, and to test how sensitive they are to the excluded proxies. I would be particularly interested to see results when the Greenland proxies are left out, since Greenland ice core d180 correlates very weakly with European temperature (e.g., Hörhold et al., 2023).

We investigated the sensitivity to choice of proxy data in Sjolte et al. (2020) with a similar method and proxy data set. This is discussed in L178-182. We do appreciate the reviewer's interest in this question and have carried out the requested test of using Greenland vs European proxies.

We made two new reconstructions based on our annual reconstruction. One based only on ice core data (RECON $_{\rm IC}$ ) and once based only on tree-ring data (RECON $_{\rm TR}$ ). We then extracted the data for tree-ring sites from RECON $_{\rm IC}$  and the ice core data from RECON $_{\rm TR}$ . The results show that the prediction of tree-ring variability from RECON $_{\rm IC}$  has weak skill, although some correlations are significant (see Figure R1-4). The best correspondence is seen for Scandinavian tree-ring records (TAA, EFmean) and winter d18O correlated to CZEC d18Ocell. Conversely, the prediction of ice core variability from RECON $_{\rm TR}$  also has weak skill, with the best correlation to GRIP d18O (0.33). This underlines the different properties of the proxy data due to differences in seasonality, region and what is recorded by the proxies. In broad terms, the main strength of the ice core data is to record large-scale atmospheric circulation for winter, while the tree-ring data capture temperature changes on a more local scale, but the wider availability of tree-ring data ensures a reasonable spatial coverage to capture a large-scale summer signal. Our main conclusion from this test is that the properties of the different sets of proxy data are complementary and constrain different seasons and regions of the reconstruction.

We will include these new figures (Figure R1-6) in the supplementary and discuss the results in section 3.1 *Ensemble reconstruction and initial evaluation* of the revised manuscript.

Minor points:

1) You should account for pattern correlation in the significance calculation; I suspect the counts of significant grid cells given in the figures is not far different than random at monthly timescales

The significance (Student's t-test) is calculated for each grid cell. The number of samples for monthly anomalies is large (8 730) and relatively low correlation correlations can be significant (p

Figure R1 European tree-ring data predicted by RECONIC.

Figure R2 Canadian tree-ring data predicted by RECONIC.

Figure R3 European d18Ocell predicted by RECONIC JJA d18O.

Figure R4 European d18Ocell predicted by RECON $_{\rm IC}$  winter d18O.

Figure R5 Greenland ice core summer d180 predicted by RECON $_{\rm TR}$ .

Figure R6 Greenland ice core winter d180 predicted by RECONTR.